# The Protective Effects of *Ganoderma lucidum* Active Peptide GLP4 on Lung Injury Induced by Cadmium Poisoning in Mice

**DOI:** 10.3390/toxics12060378

**Published:** 2024-05-22

**Authors:** Shirong Zhu, Xiaoling Wang, Gaoqiang Liu

**Affiliations:** 1Hunan Provincial Key Laboratory of Forestry Biotechnology, Central South University of Forestry & Technology, Changsha 410004, China; zsr2110574063@126.com (S.Z.); gaoliuedu@csuft.edu.cn (G.L.); 2International Cooperation Base of Science and Technology Innovation on Forest Resource Biotechnology of Hunan Province, Central South University of Forestry & Technology, Changsha 410004, China

**Keywords:** *Ganoderma lucidum* peptide, cadmium poisoning, lung injury, NLRP3 inflammasome

## Abstract

*Ganoderma* triterpenes and spore powder have shown promising results in mitigating cadmium-induced renal and hepatic injuries. *Ganoderma lucidum* active peptide GLP4 is a natural protein with dual antioxidant activities derived from the mycelium of *Ganoderma lucidum*. However, its efficacy in alleviating cadmium-induced lung injury remains unexplored. This study aims to investigate the protective effects of GLP4 against cadmium-induced lung injury in mice. Mice were exposed to cadmium chloride via nebulization to induce lung injury. The protective effect of GLP4 was assessed by measuring the total cell count in BALF, levels of inflammatory cytokines, and the expression of NLRP3 in lung tissues a through histopathological examination of lung tissue changes. The results showed that GLP4 significantly mitigated histopathological damage in lung tissues, decreased the secretion of inflammatory cytokines, and reduced the expression of NLRP3, which was elevated in cadmium-exposed mice. In vitro studies further revealed that GLP4 inhibited the cadmium-induced activation of the NLRP3 inflammasome. Notably, acute cadmium exposure by the respiratory tract did not affect the liver and kidneys of the mice. The findings suggest that GLP4 reduces cadmium-induced lung injury in mice by inhibiting the activation of the NLRP3 inflammasome, which provides a theoretical foundation for using *Ganoderma lucidum* as a preventive and therapeutic agent against cadmium poisoning.

## 1. Introduction

The escalation of human activity and increased utilization of cadmium in production have exacerbated the prevalence of environmental cadmium contamination [1,2]. Cadmium from various environmental sources, such as air, water, soil, and food, poses significant risks to human health [3]. There is a growing body of evidence suggesting that prolonged exposure to fine atmospheric particulate matter and other air pollutants increases the risk of diseases of the respiratory system [4,5]. The inhalation of pollutants, particularly cadmium, a major toxic component in atmospheric particulates, is a significant health concern [6,7,8]. Numerous studies have identified the lungs as a primary target of cadmium toxicity. Acute inhalation of cadmium fumes or aerosols can result in immediate pulmonary edema and hemorrhage, with subsequent inflammation, fibrosis, and carcinoma. Chronic exposure is associated with progressive lung fibrosis, impaired pulmonary function, and obstructive lung disease [9]. Animal studies have confirmed that acute cadmium exposure via inhalation can lead to alveolar damage, oxidative stress, apoptosis, and changes in the lung microenvironment [10,11,12,13]. Moreover, long-term cadmium exposure at low doses disrupts mitochondrial lipid metabolism in lung epithelial cells, potentially causing fibrosis [14,15]. Cellular experiments have revealed cadmium-induced inflammatory responses and apoptosis in lung epithelial cells and promoted the epithelial–mesenchymal transition (EMT) in human alveolar cells [16,17,18]. Alveolar macrophages (MHS), constituting about 70% of immune cells in the alveoli, serve as the lung’s frontline defense. They play a pivotal role in lung injury related to both infection and non-infectious factors by modulating the release of various inflammatory mediators [19]. There is substantial evidence indicating that NLRP3 inflammasomes are implicated in the inflammatory response to cadmium toxicity. Cadmium can induce renal inflammation by activating NLRP3 inflammasomes via the ROS/MAPK/NF-KappaB signaling pathway [20]. It can also trigger pyroptosis in human vascular endothelial cells, leading to cellular dysfunction [21], and exacerbate liver injury by promoting NLRP3 inflammasome activation, oxidative stress, and hepatocyte death [22]. In pigs, subacute exposure to cadmium induces lung fibrosis by activating the NF-KappaB/NLRP3 pathway, further influencing the TGFβ and PPARγ/Wnt pathways [23]. Given the critical role of NLRP3 inflammasomes in cadmium-induced injuries, further investigation is essential. Currently, there is no specific treatment for cadmium poisoning; management is primarily symptomatic, involving pain relievers, gastric lavage, and oxygen therapy. Traditional metal chelators have proven ineffective, and in combination with other drugs, they may exacerbate the condition, leading to multi-organ failure [24].

Research indicates that certain natural bioactive compounds can mitigate the adverse health effects of cadmium. These beneficial compounds include polyphenols, β-carotene, royal jelly, resveratrol, and Poria [25,26,27,28,29]. Used for over two millennia in traditional Chinese medicine, *Ganoderma lucidum* is revered for its health-enhancing and life-prolonging benefits [30]. Contemporary studies corroborate that extracts from *Ganoderma lucidum* act as natural and safe anti-inflammatory agents, effective in the prevention and treatment of inflammatory conditions [31,32]. *Ganoderma lucidum* extract is known to possess a wide range of biological activities, including anti-tumor effects, blood pressure-lowering properties, and antioxidant activity, among others [33,34,35]. In China, various herbs have traditionally been employed to manage chronic bronchitis and asthma, with recent applications of maxingshigan-yinqiaosan demonstrating promising results in preventing respiratory infections [36]. The recombinant protein rLZ-8, extracted from *Ganoderma lucidum*, demonstrates potential in alleviating OVA-induced lung inflammation by regulating the balance between Th17 and Treg cells [37]. Additionally, *Ganoderma lucidum* polysaccharides exhibit a beneficial effect in mitigating lipopolysaccharide-induced acute pneumonia by inhibiting inflammation mediated by NRP1 [38]. Our team has previously characterized the amino acid sequence of GLP4, Gln–Arg–Val–Cys–Glu, noting its potent dual antioxidant activities [39]. Building on this foundation, the present study explores the protective capabilities of GLP4 against lung injury induced by cadmium poisoning in mice. By establishing both an in vivo model of acute respiratory exposure to cadmium and an in vitro cellular injury model, we aim to elucidate the mechanisms by which GLP4 confers protection. The findings of this investigation are anticipated to lay the theoretical groundwork for employing *Ganoderma lucidum* active peptide in treating cadmium-induced pulmonary damage.

## 2. Materials and Methods

### 2.1. Reagents

The *Ganoderma lucidum* active peptide GLP4 was synthesized by Cellmano Biotech Co., Ltd., (Hefei, China). Its purity was confirmed to exceed 97% via high-performance liquid chromatography (HPLC) and mass spectrometry (MS) analyses, and it has a molecular weight of 633.72 Daltons. Cadmium chloride (CdCl_2_; Macklin, Shanghai, China) and dexamethasone (Dex; SOLARBIO, Beijing, China) were used. 

### 2.2. Animals and Treatments

Male C57BL/6J mice, aged 6–8 weeks and weighing approximately 20 ± 2 g, were procured from Hunan Slake Jingda Laboratory Animal (Changsha, China). Upon arrival, animals were accommodated with free access to standard feed and tap water, adhering to a 12 h light/dark cycle, within an environment maintained at 22 °C. All animal procedures were conducted in compliance with the Animal Care and Use Guidelines of China. The laboratory conditions were consistently controlled, with temperature set at 25 ± 2 °C and humidity set at 50 ± 5%. Throughout the experimental duration, no abnormalities in appearance, diet, or water consumption were observed, and there was no mortality among the cadmium-exposed mice. The experiment employed a respiratory aerosol exposure method. Mice were placed in a transparent sealed chamber with a capacity of 0.009 m^3^ (30 × 20 × 15 cm). An ultrasonic nebulizer (Pari, Starnberg, Germany) delivered the aerosol via a pipe connected to the chamber, with an exhaust pipe inserted below the liquid level to vent tail gases. Upon setup, the nebulization process was initiated. Thirty mice were randomly allocated into five groups (*n* = 6 per group): the Normal Control group (NC) received nebulized saline; the Cadmium Chloride model group (Cd alone) and other dosing groups were exposed to 5 mM CdCl_2_ via nebulization (2 h per day for 7 days); the Dex group (Cd + Dex) received daily intraperitoneal injections of dexamethasone (25 mg/kg) during the exposure period; the GLP4 low-dose group (Cd + GLP4_L_) received intravenous GLP4 (15 mg/kg) starting 4 days prior to Cd exposure and continuing for 7 days concurrent with Cd exposure; the GLP4 high-dose group (Cd + GLP4_H_) received intravenous GLP4 (25 mg/kg) on the same schedule as GLP4_L_. Injection volumes were kept under 0.2 mL. Mice were euthanized 24 h after final CdCl_2_ exposure for bronchoalveolar lavage fluid collection, and lung tissues were harvested for histopathology. Additionally, liver and kidney tissues were collected for biochemical and pathological assessments. The selection of dose regimens was informed by the existing literature [40,41], and preliminary trials were conducted by our research group. The experimental flow is depicted in Figure 1.

### 2.3. Bronchoalveolar Lavage Fluid (BALF) Collection and Cell Counting

The methodology for enumerating total cell counts in BALF was adapted from established protocols in the literature [42]. Briefly, within 24 h following the last exposure, thirty mice were euthanized, and a tracheal cannula was inserted. BALF was then harvested from each mouse. The procedure involved lavaging the lungs with 0.6 mL of saline, repeated three times to ensure thorough collection. The total cells retrieved from the BALF were quantified utilizing a hemocytometer.

### 2.4. Histological and Morphological Analyses

Following euthanasia, tissues from the liver, kidneys, and lungs of the mice were fixed in 4% paraformaldehyde (ECOTOP, Guangzhou, China) for 24 h at room temperature. The fixed tissues were then embedded in paraffin and sectioned into 5 μm slices. These sections were subsequently baked at 60–62 °C for 3–4 h to adhere to the slides, deparaffinized in xylene, rehydrated through a graded series of ethanol, and finally rinsed with ultrapure water. The sections were stained with hematoxylin and eosin (H&E; ECOTOP, Guangzhou, China) and the then mounted with a neutral gum sealant. The histological and morphological evaluations of the liver, kidney, and lung tissues were conducted using a Carl Zeiss Axiovert 200 microscope (Carl Zeiss, Jena, Germany).

### 2.5. Measurement of GOT and GPT Levels in Lung Tissues and BUN in Kidney Tissues, and Enzyme-Linked Immunosorbent Assay (ELISA)

The liver, kidney, and lung tissues were each homogenized to a concentration of 10% (wt/vol) in 0.9% saline solution. The activities of aspartate aminotransferase (GOT) and alanine transaminase (GPT) in the lung tissues were quantified using assay kits (Beijing Box Biotechnology, Beijing, China). Similarly, the levels of urea nitrogen (BUN) in kidney tissues were measured using a kit (SOLARBIO, Beijing, China). Furthermore, the supernatants from cell cultures and BALF were collected for the quantification of inflammatory markers. These levels were determined utilizing an ELISA kit in accordance with the instructions of the manufacturer (RENJIEBIO, Shanghai, China).

### 2.6. Immunohistochemistry

For antigen retrieval, the sections were microwaved for 10 min. Subsequently, a histochemical marker pen was used to delineate the tissue sections before applying 3% hydrogen peroxide for 20 min at 37 °C, followed by a PBS wash. The sections were then blocked with 10% goat serum at room temperature for 30 min. Following this, the sections were incubated overnight at 4 °C in a humidified chamber with anti-NLRP3 primary antibody (1:800, Cell Signaling Technology, Boston, MA, USA). Post-primary-antibody incubation, the sections were washed with PBS and treated with HRP-conjugated universal secondary antibodies for 2 h at 37 °C. Another round of PBS washes preceded the addition of fresh DAB for chromogenic development. Hematoxylin was used for counterstaining, after which the slides underwent dehydration and mounting. Microscopic examination was performed to capture images of the stained tissues.

### 2.7. Cell Culture and Treatment

Mouse alveolar macrophages (MHS) were obtained from Pricella Biotech (Wuhan, China) and maintained in RPMI-1640 medium enriched with 0.05 mM β-mercaptoethanol, 10% fetal bovine serum (FBS), and 1% penicillin–streptomycin. The cells were cultured under sterile conditions at 37 °C in a 5% CO_2_ atmosphere. For the treatment studies, the MHS cells were grouped as follows: untreated control (control), cadmium chloride-treated (Cd alone), Dex-treated positive control (Cd + Dex), and GLP4-treated (Cd + G-12.5/G-25). The cells were plated in 6-well plates and allowed to adhere until they occupied 80% of the surface area. Following the removal of the original medium, the cells were pre-treated with medium containing either GLP4 at concentrations of 12.5 or 25 mg/mL or dexamethasone at a concentration of 6 μM for 3 h. Cadmium chloride was then added to achieve a final concentration of 50 μM, and incubation was continued for an additional 6 h. Post-treatment, cells were rinsed twice with PBS. Cell lysates were collected for Western blot and RT-PCR analyses, while supernatants were preserved at −80 °C for the subsequent measurement of pro-inflammatory cytokines.

### 2.8. Cell Viability Assay

The viability of MHS cells after GLP4 treatment was assessed using a CCK-8 assay kit (ECOTOP, Guangzhou, China). MHS cells were seeded at a density of 6000 cells per well in 96-well plates. The cells were treated with various concentrations of GLP4 (ranging from 0 to 200 mg/mL) for 24 h. Following this treatment period, 10 µL of CCK-8 solution was added to each well and incubated for 2 h to facilitate the reaction. The optical density at 450 nm was then measured using a microplate reader (Thermo Fisher Scientific, Waltham, MA, USA).

### 2.9. Quantitative Real-Time PCR

Total RNA was isolated from MHS cells using an extraction kit (ECOTOP, Guangzhou, China). The RNA pellets were resuspended in 50 µL of diethylpyrocarbonate (DEPC)-treated water. The concentration of total RNA was ascertained via spectrophotometry at 260 nm. Reverse transcription of RNA to cDNA was performed using the RT reagent kit with gDNA Eraser (YEASEN, Shanghai, China). Subsequent cDNA amplification was carried out with Hieff Qpcr SYBR Green Master Mix (YEASEN, Shanghai, China) using a real-time PCR detection system. Relative gene expression levels were quantified using the 2^−ΔΔCT^ method. GAPDH were used for gene expression normalization in cells. Specific primers for NLRP3, IL-1β, IL-6, TNF-α, and GAPDH were designed using the Primer-Blast tool from the National Center for Biotechnology Information (NCBI, USA). Details of the primer sequences are listed in Table 1.

### 2.10. Western Blot Assay

MHS cells were lysed for 35 min using RIPA buffer (Beyotime, Shanghai, China) with 1 mmol/L phenylmethanesulfonyl fluoride (PMSF) (MilliporeSigma, Billerica, MA, USA) and a protease inhibitor cocktail (Servicebio, Wuhan, China). The samples were centrifuged at 12,000× *g* for 15 min at 4 °C, and protein levels in the supernatants were determined using a BCA protein assay kit (Beyotime, Shanghai, China). An equal quantity of protein (10 µg) from each sample was separated via 10% SDS-PAGE and subsequently transferred onto PVDF membranes. The membranes were incubated in 5% non-fat dry milk dissolved in Tris-buffered saline with 0.1% Tween-20 (TBST) to block non-specific binding sites for one hour and then incubated with primary antibodies at a dilution ratio of 1:1000 overnight at 4 °C. After washing, the membranes were exposed to secondary antibodies (1:10,000) for 1 h at room temperature. Primary antibodies IL-1β (ab283818, 1:1000), IL-18 (ab207323, 1:1000), caspase-1 (ab179515, 1:1000), and ASC (ab307560, 1:1000) were obtained from Abcam (Cambridge, UK). β-actin (EM21002, 1:1000) was acquired from HUABIO (Hangzhou, China), and NLRP3 (15,101, 1:1000) was purchased from CST (Boston, MA, USA). HRP goat anti-rabbit IgG (1:10,000) and HRP goat anti-mouse IgG (1:10,000) were obtained from antgene (Wuhan, China). Ultimately, the blots were visualized using a chemiluminescence detection system (Advansta, Menlo Park, CA, USA).

### 2.11. Statistical Analysis

Statistical analyses were conducted using GraphPad Prism software, version 9.0.0 (GraphPad Software, La Jolla, CA, USA). The data are reported as means ± standard error of the mean (SEM). One-way analysis of variance (ANOVA) with Tukey’s multiple comparisons test was employed as suitable for the analysis. The threshold for statistical significance was set at *p* < 0.05.

## 3. Results

### 3.1. Protection by GLP4 against Cadmium Poisoning-Induced Lung Injury in Mice

The efficacy of GLP4 in counteracting lung injury caused by cadmium poisoning was investigated in vivo. Control mice exhibited vigor, normal activity, robust feeding behavior, and glossy fur. In contrast, mice subjected to cadmium exhibited lethargy, irritability, diminished appetite, reduced mobility, and scruffy fur, along with a decrease in body weight (Figure 2A). Elevated counts of total cells and increased levels of inflammatory markers IL-1β, IL-6, and TNF-α were detected in the BALF of the cadmium-exposed mice (Figure 2B,C). Histopathological examination revealed that cadmium intoxication led to the accumulation of inflammatory cells and disruption of the alveolar architecture, with diffuse damage across alveoli, alveolar sacs, alveolar septa, and bronchi (Figure 3). These findings demonstrate that the inflammatory response and pathological damage in the lungs are exacerbated by cadmium poisoning. The innate immune system, particularly the NLRP3 inflammasome, is crucial in the body’s response to cadmium-induced injury and is a focal point in the study of cadmium toxicity [21,22,43,44]. The role of the NLRP3 inflammasome in cadmium-induced lung injury was evaluated by immunohistochemically quantifying NLRP3 levels in lung tissue (Figure 4). Exposure to cadmium upregulated NLRP3 expression in the lung tissues, triggering the activation of the NLRP3 inflammasome. This activation promotes the autocleavage of caspase-1 precursors, which in turn facilitates the maturation of proinflammatory cytokines IL-1β and IL-18, contributing to the inflammatory process. Elevated NLRP3 expression suggests that cadmium poisoning may trigger the activation of the NLRP3 inflammasome, contributing to lung injury in mice. Notably, GLP4 administration resulted in a reduction in both the total cell count and the levels of inflammatory markers in the BALF of cadmium-exposed mice (Figure 2B,C). Moreover, GLP4 attenuated the detrimental impact of cadmium on lung tissue structure and inflammatory cell infiltration (Figure 3). Correspondingly, there was a decrease in NLRP3 levels within the lung tissues (Figure 4). Collectively, these findings indicate that GLP4 exerts a protective effect against lung injury induced by cadmium poisoning in mice.

### 3.2. Respiratory Exposure to Acute Cadmium Does Not Affect Mouse Liver and Kidney Function

We evaluated the potential damage to mouse liver and kidney function through biochemical indices and pathological assessments. Our results indicated that acute inhalation of cadmium neither altered the levels of glutamic-oxaloacetic transaminase (GOT) and glutamic-pyruvic transaminase (GPT) in the liver nor affected the blood urea nitrogen (BUN) levels in the kidneys when compared with the control group (see Figure 5A–C). Additionally, pathological examinations revealed no significant damage in these organs. Liver sections demonstrated an intact lobular architecture, with hepatocytes arranged in orderly rows, consistent in size, possessing abundant cytoplasm and clearly delineated nuclei (illustrated in Figure 5D). Similarly, kidney sections displayed structurally intact and uniformly shaped glomeruli without separation from the capsule wall. The renal tubular cells were compactly organized, with uniformly sized nuclei that were readily visible (depicted in Figure 5E). Collectively, these observations indicate that acute cadmium inhalation does not adversely impact hepatic or renal functions in mice.

### 3.3. GLP4 Mitigates the Inflammatory Response in MHS Cells Induced by Cadmium by Suppressing NLRP3 Inflammasome Activation

NLRP3 is implicated in the cadmium-induced polarization of pulmonary macrophages in porcine models [23]. Previous studies, such as those documented in Section 3.1, have demonstrated that cadmium exposure leads to the upregulation of NLRP3 expression in murine lung tissue. Given these findings, we postulated that GLP4 may mitigate the inflammatory response elicited by cadmium through the inhibition of NLRP3 inflammasome activation in an in vitro setting. Previous research has established that cadmium can provoke a pro-inflammatory phenotype in MHS cells, contributing to the exacerbation of pulmonary damage [45]. In this context, we employed MHS cells as a model to assess the impact of GLP4 on NLRP3 activation and the consequent production of the pro-inflammatory cytokines IL-1β, IL-6, and TNF-α. Upon exposure to cadmium, MHS cells exhibited an upsurge in the expression of IL-1β, IL-6, and TNF-α compared with control cells (Figure 6B–D). Additionally, there was a marked increase in the mRNA levels of IL-1β, TNF-α, and NLRP3 (Figure 6E–G), as well as the protein levels of NLRP3 inflammasome-associated proteins in the cadmium-treated cells (Figure 7A–F). Intriguingly, the administration of GLP4 significantly mitigated these effects. Taken together, our results strongly suggest that GLP4 may ameliorate cadmium-induced inflammatory responses by impeding the activation of the NLRP3 inflammasome.

## 4. Discussion

The mechanism of cadmium (Cd) annihilation is closely associated with its behavior in the environment and biological systems, particularly its toxicity and metabolic pathways. Within living organisms, cadmium can form complexes by binding to various biomolecules, including metallothioneins. The elimination of cadmium occurs through multiple routes, such as urine, feces, and sweat. The kidneys play a crucial role in excreting cadmium through urine, eliminating a portion of the absorbed cadmium. Additionally, the intestines contribute to cadmium elimination by excreting unabsorbed cadmium through feces. Although cadmium itself is not biodegradable, it can undergo transformations into different chemical forms facilitated by microorganisms and enzymes, thereby reducing its toxicity. For instance, certain microorganisms possess the ability to convert cadmium ions into various valence states or compounds through reduction or oxidation processes, altering their bioavailability and toxicity [46]. Cadmium toxicity is predominantly harmful to organisms by inducing apoptosis, disrupting metal element metabolism, causing mitochondrial damage, and promoting oncogene expression. Oxidative stress and inflammation are key mechanisms underlying the adverse effects of cadmium exposure [47]. In natural settings, water mist can interact with atmospheric cadmium in windy conditions, leading to the formation of water-soluble cadmium-containing aerosols. Among the techniques for liquid nebulization—mechanical, pneumatic, and ultrasonic—pneumatic nebulization is most commonly employed in studies of inhalation toxicology [40]. In our study, we utilized pneumatic nebulization of an aqueous cadmium solution to replicate the respiratory pathway of cadmium exposure and established a mouse model for lung injury induced by cadmium poisoning. Following exposure to cadmium, mice exhibited significant health declines, including weight loss, reduced appetite, behavioral changes indicative of depression, and a deterioration in the condition of their fur [40]. Furthermore, an increase in cytokines TNF-α, IL-1β, and IL-6 was observed in BALF [45]. highlighting the extensive damage cadmium poisoning inflicts on the mice’s overall health. Interestingly, active compounds found in *Ganoderma lucidum* have shown protective effects against cadmium poisoning. Specifically, *Ganoderma lucidum* spore powder and polysaccharides demonstrated a protective capability against liver and kidney injury induced by cadmium poisoning, respectively [48,49], Additionally, triterpenoids from *Ganoderma lucidum* were effective in mitigating oxidative stress and inflammatory damage in chicken liver caused by cadmium [50].

Recent research has largely overlooked the impact of *Ganoderma lucidum* polypeptides on cadmium poisoning. In our study, we specifically examined the effect of GLP4 on cadmium-induced toxicity. We found that GLP4 notably alleviated histopathological changes in the lungs, pulmonary edema, the production of inflammatory cytokines, and the activation of the NLRP3 inflammasome. Consequently, GLP4 significantly reduced lung injury in mice resulting from cadmium poisoning. While extensive experimental work has explored the consequences of dietary and intraperitoneal cadmium chloride exposure on liver and kidney damage in mice [51,52,53], research into acute cadmium exposure via the respiratory route leading to hepatic and renal injury is less common. Our findings indicate that acute respiratory exposure to cadmium does not adversely affect the liver or kidneys, as evidenced by histopathological examinations and biochemical tests. This observation suggests that the respiratory pathway of cadmium exposure may have a distinct impact on organ toxicity compared with other routes of exposure. Moreover, the protective effects of *Ganoderma lucidum* peptide GLP4 against lung injury highlight its potential as a therapeutic agent in mitigating the harmful effects of cadmium, emphasizing the need for further investigation into its benefits across different exposure pathways.

Cadmium has been identified as a promoter of inflammatory responses in MHS cells, with its exposure leading to a time-dependent increase in the expression of inflammatory factors. This exacerbates lung injury by engaging the innate immune system’s first line of defense [45]. The NLRP3 inflammasome, a critical sensor in this system, forms in response to both exogenous pathogens and endogenous cellular damage. It comprises NLRP3, which detects danger and recruits molecules; caspase-1, essential for cytokine maturation and processing gasdermin D for cytokine release and sepsis; and ASC, bridging NLRP3 and caspase-1 [54]. Emerging research suggests Ganoderma acid A’s potential in modulating neuroimmunity and enhancing antidepressant behavior by inhibiting NLRP3 inflammasome activity [55]. Additionally, *Ganoderma lucidum* polysaccharide GLPS improves liver function in injured mice by reducing NLRP3 inflammasome activation [56], illustrating its therapeutic prospects in modulating immune responses.

To date, research on the interaction between *Ganoderma lucidum* proteins and NLRP3 inflammasome activation remains limited. This study demonstrates that cadmium exposure activates the NLRP3 inflammasome in alveolar macrophages, leading to an increased expression of pro-inflammatory factors IL-1β, TNF-α, and IL-6. Interestingly, administration of GLP4 effectively reduced the expression of the NLRP3 inflammasome at both gene and protein levels, offering protection against cadmium-induced lung injury. Despite these findings, the study did not confirm the causal link between cadmium exposure and NLRP3 inflammasome activation through reverse validation using an NLRP3 inhibitor. Therefore, while it is hypothesized that cadmium-induced lung injury may be associated with NLRP3 inflammasome activation, further investigation is warranted. This research contributes to the theoretical framework for studying the inhibitory effects of GLP4 on NLRP3 inflammasome activation.

## 5. Conclusions

In conclusion, the results of this study reveal that GLP4, an activated peptide from *Ganoderma lucidum*, has the potential to mitigate lung injury induced by cadmium exposure through the inhibition of NLRP3 inflammasome activation. It is noteworthy that acute respiratory exposure to cadmium does not impact liver or kidney function in mice. These findings highlight the significance of GLP4 as a promising candidate for the development of natural therapeutics aimed at preventing and treating lung injuries caused by cadmium poisoning. This study underscores the therapeutic potential of *Ganoderma lucidum* derivatives in addressing environmental health challenges and offers a foundation for further exploration into their beneficial effects. However, based on the available data, we were unable to determine the impact of GLP4 on the downstream signaling pathway associated with NLRP3 inflammasome activation in acute cadmium exposure via the respiratory route. Future studies will focus on investigating the protective effect of GLP4 on cadmium-induced lung injury in mice through chronic and subacute cadmium exposure via the respiratory route. Additionally, further research is required to elucidate the molecular mechanism by which GLP4 inhibits NLRP3 inflammasome activation.

## Figures and Tables

**Figure 1 toxics-12-00378-f001:**
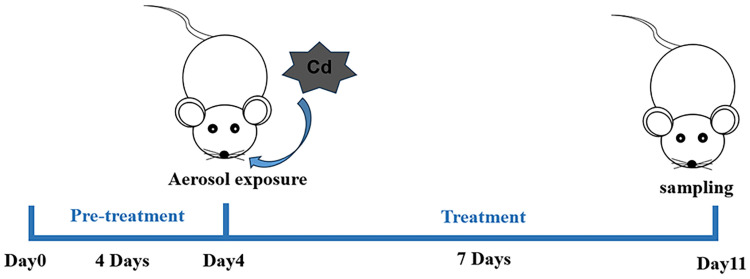
Animal experimental design. Mice were subjected to daily intravenous injections of GLP4 (doses of 15 mg/kg and 25 mg/kg; a volume < 0.2 mL per mouse) via the tail vein for four consecutive days. Following this, they were exposed to a CdCl_2_ aqueous aerosol for a duration of seven days. As a control, the NC (normal control) group was exposed to an equivalent volume of saline. For positive control purposes, mice received intraperitoneal injections of dexamethasone (25 mg/kg/day; volume < 0.2 mL per injection) concurrent with cadmium exposure. On the eleventh day, euthanasia was carried out using isoflurane, after which BALF (bronchoalveolar lavage fluid) was collected, and samples of lung, liver, and kidney tissues were harvested for analysis.

**Figure 2 toxics-12-00378-f002:**
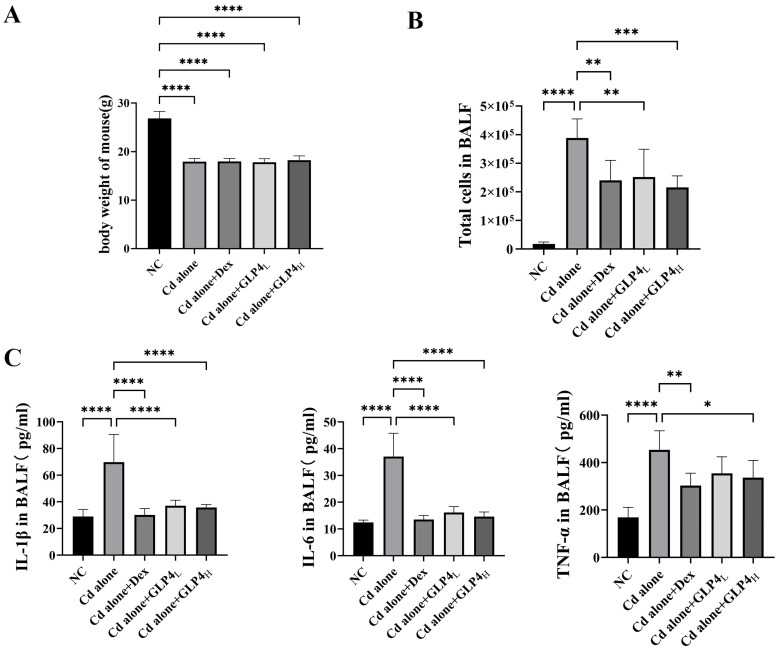
Reduction in cadmium-induced lung injury by GLP4. (**A**) Body weight measurements. (**B**) Total cell counts in bronchoalveolar lavage fluids (BALFs). (**C**) Cytokines (IL-1β, IL-6, and TNF-α) in BALFs measured via ELISA (*n* = 6). Statistical significance is denoted by * *p* < 0.05, ** *p* < 0.05, *** *p* < 0.0005, and **** *p* < 0.0001, as determined through one-way ANOVA.

**Figure 3 toxics-12-00378-f003:**
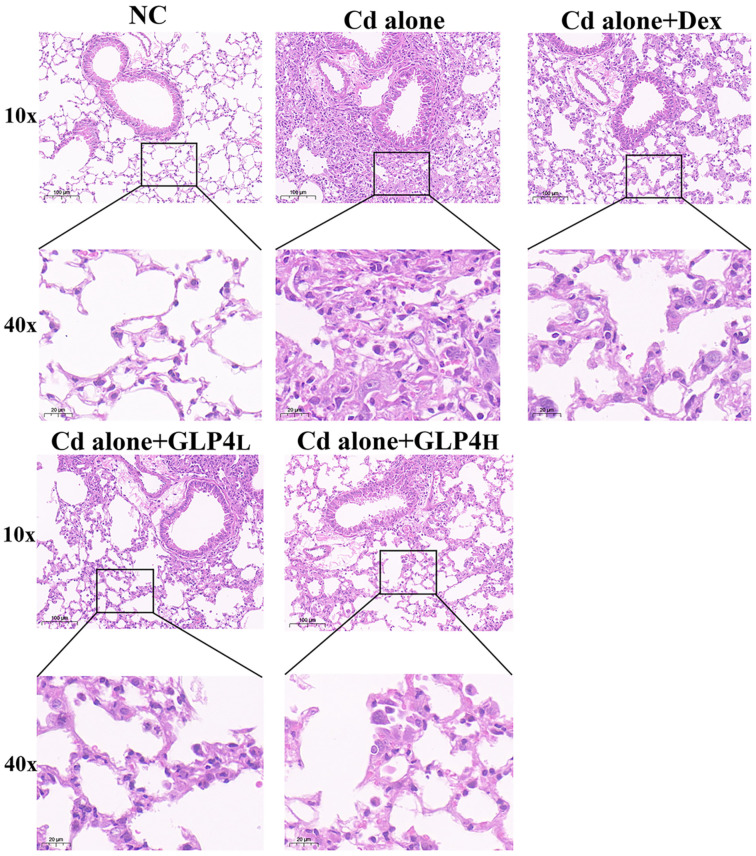
Reduction in cadmium-induced lung injury by GLP4. Representative results of lung tissue samples were stained with hematoxylin and eosin, Bar = 100 µm and 20 μm, and magnification = 100× and 400×, respectively.

**Figure 4 toxics-12-00378-f004:**
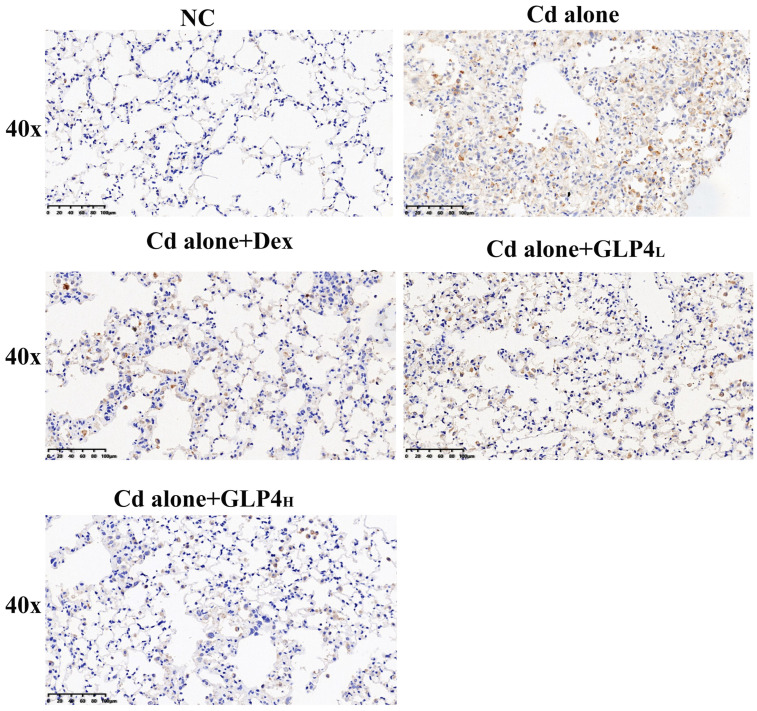
GLP4 reduces NLRP3 protein expression in cadmium-induced lung tissue. Immuno-histochemistry NLRP3 protein expression levels in the mouse lung tissues were examined; Bar = 100 μm; magnification 400×.

**Figure 5 toxics-12-00378-f005:**
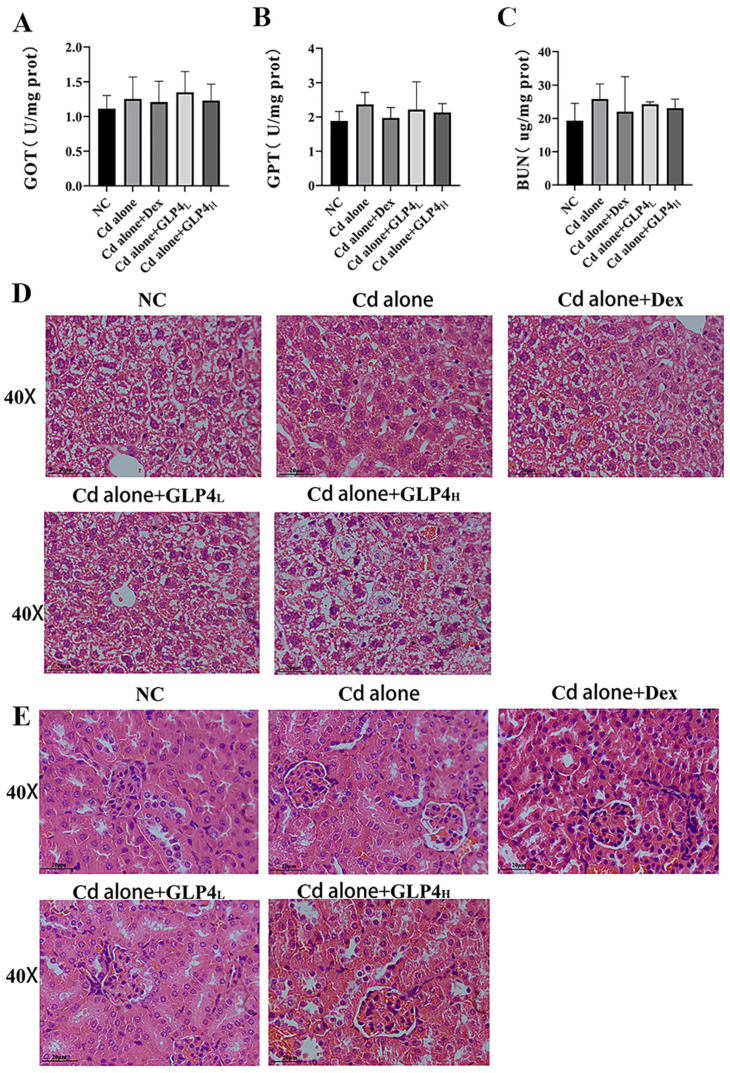
No impact on liver and kidney function in mice after acute respiratory exposure to cadmium. (**A**,**B**) Levels of glutamic-pyruvic transaminase (GOT) and glutamic-oxaloacetic transaminase (GPT) in mouse liver tissue (*n* = 6). No significant difference. (**C**) Blood urea nitro-gen (BUN) levels in mouse kidney tissues (*n* = 6). No significant difference. (**D**) Representative results of livers tissue samples stained with hematoxylin and eosin, magnification 400×, Bar = 20 μm. (**E**) Representative results of kidneys tissue samples stained with hematoxylin and eosin, magnification 400×, Bar = 20 μm.

**Figure 6 toxics-12-00378-f006:**
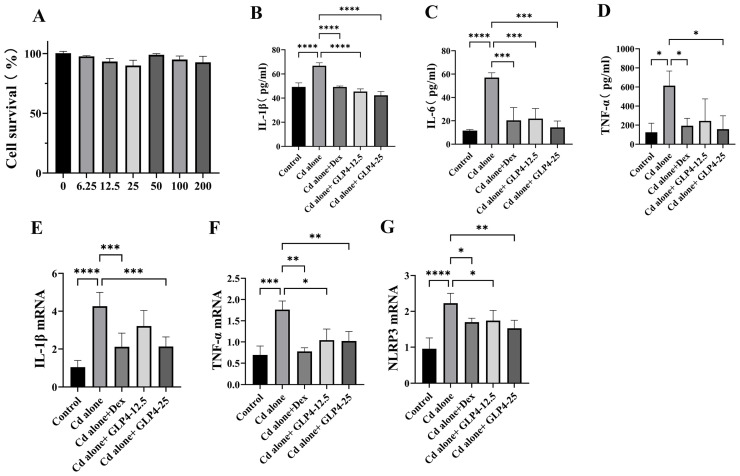
GLP4 diminishing the cadmium-induced expression of inflammatory cytokines in vitro. (**A**) Impact of GLP4 on the viability of MHS cells. (**B**–**D**) Cytokines (IL-1β, IL-6, and TNF-α) in MHS cells measured via ELISA (*n* = 3). (**E**–**G**) mRNA (IL-1β, TNF-α, and NLRP3) in MHS cells measured via RT-PCR (*n* = 3). Data represent the mean ± SD for *n* = 3 samples per group. Statistical significance is denoted by * *p* < 0.05, ** *p* < 0.05, *** *p* < 0.0005, and **** *p* < 0.0001, as determined through one-way ANOVA.

**Figure 7 toxics-12-00378-f007:**
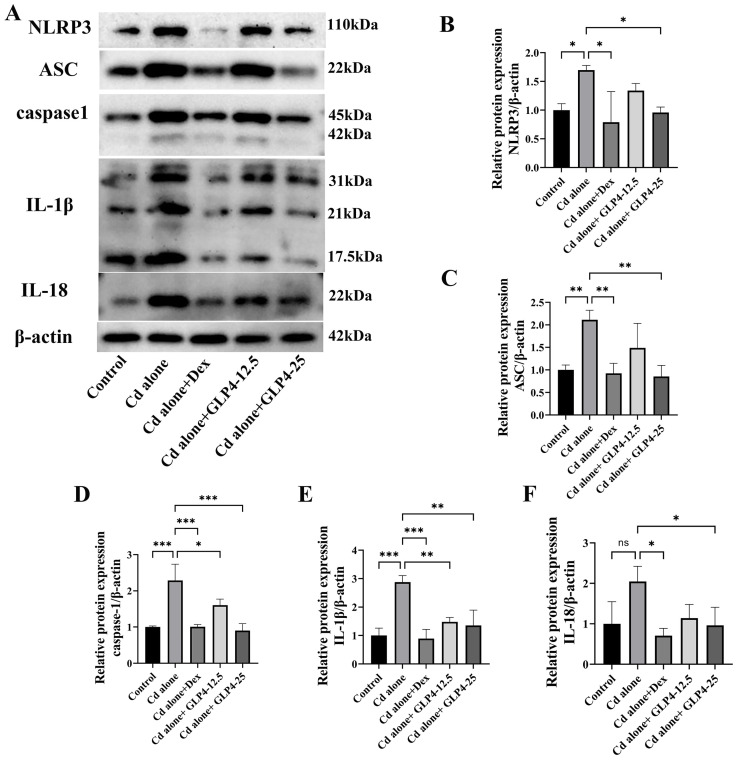
GLP4 suppression of cadmium-induced NLRP3 inflammasome activation in vitro leads to diminished expression of inflammatory cytokines. (**A**–**F**) The effect of CdCl_2_ exposure on the NLRP3 inflammasome in MHS cells, investigated and quantitatively analyzed using Western blot (*n* = 3). Data represent the mean ± SD for *n* = 3 samples per group. Statistical significance is denoted by * *p* < 0.05, ** *p* < 0.05, and *** *p* < 0.0005, as determined through one-way ANOVA.

**Table 1 toxics-12-00378-t001:** Primers sequences used for real-time PCR reactions.

Genes	Primer Pairs
NLRP3	Upper: AGGCTGCTATCTGGAGGAACT Lower: CCTTTCGGGCGGGTAATC
IL-1β	Upper: GAAATGCCACCTTTTGACAGTG Lower: TGGATGCTCTCATCAGGACAG
IL-6	Upper: TAGTCCTTCCTACCCCAATTTCC Lower: TTGGTCCTTAGCCACTCCTTC
TNF-α	Upper: CAGGCGGTGCCTATGTCTC Lower: CGATCACCCCGAAGTTCAGTAG
GAPDH	Upper: CATCACTGCCACCCAGAAGACTG Lower: ATGCCAGTGAGCTTCCCGTTCAG

## Data Availability

All relevant data are within this paper.

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
