# Peer review of "The Protective Effects of Ganoderma lucidum Active Peptide GLP4 on Lung Injury Induced by Cadmium Poisoning in Mice"

_toxics, 2024, doi:10.3390/toxics12060378_

Round 1

Reviewer 1 Report

Comments and Suggestions for Authors

In the text below I am giving my comments on manuscript entitled “The Protective Effects of Ganoderma Lucidum Active Peptide GLP4 on Lung Injury Induced by Cadmium Poisoning in Mice”.

To my opinion, the manuscript is well written on sufficiently good English. The experimental design is logically structured and the obtained results fully explain the aim of the study.

I would advise the authors to present their figures in better resolution or placed on one page, since they are very large. Another point is whether the authors are using a positive control in order to compare the activity of the investigated peptide and if not what their explanation is.

Comments on the Quality of English Language

Minor editing of English language required

Reviewer 2 Report

Comments and Suggestions for Authors

The introduction is well documented, and the studied bibliography is recent and highlights the relevant elements for the research. These are very recent data, and the research is in good correlation with them.

The research project responds to current needs in the conditions where heavy metal pollution to which we are involuntarily exposed due to its presence in the air, proves to have disastrous effects on human health.

The methods used by the research group are adequate and allow the achievement of the project's objectives

The results are well highlighted and presented in such a way that they can be easily correlated with the methodology used. 

It is interesting to know, in order to be able to evaluate the degree of danger, from which level of the Cd dose the symptoms start to appear. If the authors have the data, I recommend that they present it.

Also, a short and simple presentation of the cadmium annihilation mechanism.

The conclusions of the study are relevant, even if they briefly present research results

I recommend that the conclusions specify the fact that acute respiratory exposure to cadmium does not affect the liver and kidney function of mice.

Reviewer 3 Report

Comments and Suggestions for Authors

Dear Authors,

This Manuscript addresses the protective properties of GLP4 on lungs damaged by cadmium poisoning in mice. However, the Article requires the necessary following revisions before publication. 

line 200 - provide details of the antibodies used

Fig. 1 - too illegible, too small font (graphs), too small magnification (photos).

On the microscope photos (Fig. 1E, 1F, 2D, 2E), please include the scale and magnifications of the specific photo. Also, under these photos, please indicate the staining technique used.

Fig. 3 - too illegible, too small font.

Please expand the "Conclusion" section - w please provide a proposal for future research that is being considered by the Authors or other researchers. 

Please make the Introduction more interesting - when reading it, the reader feels insufficient information regarding the source of the peptide, i.e. Ganoderma lucidum. The authors mention only anti-inflammatory properties. Am I to understand that others this plant does not have? Does G. lucidum perhaps have any proven activity in respiratory infections? Maybe there are some reports that talk about its effect on bacteria associated with pneumonia? If possible, please add this information that will reinforce the importance of the topic taken up in this Manuscript. 

Best regards
